

# The sensitivity of Landscape Evolution Models to DEM grid cell size

Christopher J Skinner[1], Thomas J Coulthard[1]

[1]Energy and Environment Institute, University of Hull, UK

*Correspondence to*: Chris J Skinner (C.Skinner@hull.ac.uk)

**Abstract.**

Landscape evolution models (LEMs) are useful for understanding how large scale processes and perturbations influence the development of planetary surfaces. With their increasing sophistication and improvements in computational power they are finding greater uptake in analyses at finer spatial and temporal scales, however. For many LEMs, the planetary surface is

represented by a grid of regularly spaced and sized grid cells, or pixels, referred to as a Digital Elevation Model (DEM), yet despite the importance of the DEM to LEM studies there has been little work to understand the influence of grid cell size (i.e. resolution) on model behaviour and outputs. This is despite the choice of grid cell size being arbitrary for many studies, with users needing to balance detail with computational efficiency. Using the global sensitivity analysis Morris Method, the sensitivity of the CAESAR-Lisflood LEM to the DEM grid cell size is evaluated relative to a set of key user-defined

parameters, showing it had a similar level of influence as a key hydrological parameter and the choice of sediment transport law. Outputs relating to discharge and sediment yields remained stable across different grid cell sizes until the cells became so large that the representation of the hydrological network degraded. Although total sediment yields remained steady when changing the grid cell sizes, closer analysis revealed that using larger grid resulted in it being built up from fewer yet more geomorphically-active events, risking outputs that are 'the right answer but for the wrong reasons". These results are important

considerations for modellers using LEMs and the methodologies detailed provide solutions to understanding the impacts of modelling choices on outputs.

## 1 Introduction

Landscape evolution models simulate the morphodynamic change of landscapes typically over long time scales ranging from decades to multi-millennia (van der Beek, 2013). Landscape change is often simulated by applying process based rules of

hydrology, erosion and deposition to change the elevation of cells in a regular grid, or points in an irregular mesh, that represents the land surface. The spatial resolution of this virtual surface is an important consideration due to two contrasting effects. Firstly, if it is too coarse (e.g. larger grid cells) it may smooth out the terrain too much and miss out key landscape features. Secondly, a finer of higher spatial resolution will better represent features but increases the number of cells and points for the area simulated that in turn increases the computation time (halving the grid cell size results in a square increase in the





number of grid cells). Therefore a compromise DEM grid cell size is used by LEMs that captures drainage basin and hillslope features whilst maintaining a low number of grid cells (Hancock, 2005; Hancock et al., 2016).

Unexpectedly, in Landscape Evolution Modelling, there are few studies that specifically address the impacts of DEM resolution. Though Schoorl et al., (2000) used the LAPSUS model to simulate landscape development on a series of artificial

DEMs of simple slopes and catchments with resolutions of 1, 3, 9, 37 and 81m respectively. Schoorl et al., (2000) results showed that with larger grid cells total erosion or sediment yield from the simulations increased and that this was due to an in increase in erosion coupled decrease in sedimentation. They argued that the erosion increase was due to the model parameterisation, but that a decrease in the physical representation of the landscape with larger grid cells made sedimentation more difficult, concluding that it is important that the extent of the landscape and its relief characteristics are realistically

represented by the used DEM. Pelletier, (2010) noted an impact of grid cell size in LEM's where using larger grid cells flow paths can become dominated by only being able to change direction by 45 or 90 degrees.

The lack of DEM resolution studies is someone surprising, considering there is research indicating the sensitivity of LEM's to the DEM used. For example, Hancock, (2006) showed a sensitivity in LEM outputs to DEM's created with different

kriging/interpolation methods. These changes in the representation can then have important cumulative impacts if the landscape is modelled as Landscape Evolution Models may exacerbate or deepen concavities or other features ultimately leading to different shape topographies (Ijjasz-Vasquez et al., 1992; Willgoose et al., 2003). Hancock et al., (2016) illustrated this by perturbating a DEM by different ranges of random values and simulating millennial timescale changes on the different surfaces using the SIBERIA LEM. They found that an increasing magnitude of random surface variability did not significantly

alter total basin sediment yields, but greatly changed the temporal pattern or delivery of sediment output. Furthermore, after 10 000 years of simulation the alternative positions of initial random perturbations strongly influenced local patterns of hillslope erosion and landscape evolution - although general landscape metrics were very similar. Hancock and Evans (2006) looked at two small catchments in North Australia using 10, 20, 30, 40 and 50m grid cells to evaluate the impact of resoltuion in determining channel head location and the area–slope relationship and cumulative area distribution that is a key driver in

the SIBERIA (Willgoose and Riley, 1998) LEM. Their findings showed a clear drop in the slope/area relationship with larger grid cells – due largely to the smoothing and subsequent simplification of topography. Finally, Finlayson and Montgomery (2003) show a major degradation of DEM mean slope values when resampling from 30 to 90 to 900m – representing the smoothing of features.

In cellular morphodynamic models (similar in many ways to LEMs) Doeschl-Wilson and Ashmore (2005), examined the Murray and Paola (1994) braided river model and noted that the model performance was strongly affected by the spatial scales at which the input topography were represented. They demonstrated that when tested over a range of different spatial resolutions the model had a 'preferred' scale where it self-adjusted to have a channel width with a certain number of cells



(rather than a distance represented by a number of cells) (Doeschl-Wilson and Ashmore, 2005). Possible reasons why there is

a sensitivity to grid resolution in cellular approaches was discussed by Nicholas (2005), who stated that this was a consequence of the water and sediment routing equations used in simplified cellular models. For example, where sediment and water were routed in proportion to local bed slopes, the calculations may become sensitive to very small variations in elevation as grid cell resolution changes (Nicholas 2005), that also shows a weakness in using local bed slope to represent the energy slope. This is especially important in a LEM or morphodynamic model where these elevations will be changing every iteration in

response to erosion and deposition – this effect will be amplified or reduced by grid resolution.

The two dimensional flow of water over landscapes is a key process in LEMs and for two dimensional hydraulic models of flood inundation the effects of DEM resolution have been extensively studied (e.g. (Horritt and Bates, 2001; Savage et al., 2016). Horritt and Bates, (2001) tested the LISFLOOD-FP inundation model against satellite derived flood inundation extents

over DEMs with gird cell resolutions ranging from 10 to 1000m. Overall, they showed a good comparison between inundation area/extent over all resolutions (using the same model calibrations) though comparison of flood wave travel time was notably different. Interestingly, this shows how DEM resolution was less important in spatial matches between observed and modelled water extents, but certainly interfered with the equations determining where water went (travel times) in effect simplifying them to a point where they did not perform adequately with respect to resolution. Claessens et al., (2005) summarise these

effects neatly: that the DEM resolution acts to firstly simplify the topographic data, and secondly any model processes or governing equations that operate below this resolution will therefore also be simplified. This can lead to apparent gains in accuracy due to greater process representation within the model being countered by the coarser model resolution (Claessens et al., 2005). Horritt and Bates, (2001) also described how changes in topographic detail with different resolution DEMs also affected floodplain storage. Similar topographic degradation affecting model behaviour was observed by Savage et al., (2016)

where using LISFLOOD-FP to simulate inundation over a wide range of resolutions they noted that model performance degraded where grid cells were larger than 50m. This was due to the channel being poorly represented within the DEM leading to increased floodplain water depths – lower velocities that all affected negatively model performance. Importantly, Savage et al., (2016) also observed how model resolution affected *parameter sensitivity* a secondary affect aside from model performance. This was also a key finding of Lim and Brandt, (2019) using the hydraulic component of CAESAR-Lisflood

LEM to examine any dependency between DEM resolution, Manning's *n* roughness coefficient*,* and model performance. Comparing model inundation extents and depths for flood events on two rivers to simulation results over DEM resolutions from 1 – 50m, they demonstrated that high-resolution DEMs performed better with higher Manning's *n* values whereas lower *n* values gave better outputs for lower resolution DEMs. Lim and Brandt, (2019) also showed that whilst coarser resolution DEMs generated better value performances according to their metrics, there were more discrepancies between known flooding

and predicted water surface elevations illustrating a dependency on the metric used for assessment. Choice of metric for assessing model performance is also an important issue presently facing LEM studies (Skinner et al., 2018).





In Computational Fluid Dynamics (CFD) where more complex numerical methods are used for hydraulic modelling, the effects of different grid resolutions or meshing methods are widely considered. Where CFD model simulations are applied to

engineering solutions there are controls and standards for the verification of models (Vassiliadis et al., 2001) that are also reflected in the journal publication policies such as "*Solutions over a range of significantly different grid resolutions should be presented to demonstrate grid independent or grid-convergent results*" (Roache, 2019; Roache et al., 2009). Here grid independent (or grid independence) refers to whether errors or differences between different resolution simulations are sufficiently small. Hardy et al., (2003) provide a clear summary and example of methods for assessing grid independence using

a 'Grid Convergence Index approach'. Nicholas (2005) comments that whilst grid-independence is considered a key requirement of computational fluid dynamics (CFD) approaches – it may not be reasonable to use such approaches in cellular methods. A logical step might be to use methods from CFD grid independence testing on LEM models. However, grid independence tests are largely during steady flow conditions (e.g. Hardy et al., (2003)) measuring flow velocities in x, y, and z directions (for example) but sediment transport in LEM and morphodynamic models is highly episodic and non-linear even

when averaged over medium time scales (Coulthard et al., 2010; Coulthard and Wiel, 2012). Therefore, the availability and choice of metrics to assess LEM performance is difficult.

This issue of which metrics to use to assess LEM model performance was considered by (Skinner et al., 2018) where they carried out a multidimensional sensitivity analysis on the CAESAR-Lisflood (Coulthard et al., 2013) LEM. Previously, such

studies have been hampered by long model run times making Monte-Carlo style analyses difficult, but here Skinner et al., (2018) used the Morris Method (Morris, 1991) to analyse the sensitivity of 15 different model parameters on model performance (Figure 1). Key to this study was the assessment model behaviours across 15 model functions across four core behaviour groups: catchment sediment yields; internal geomorphology; catchment discharge; and, model efficiency.

Usefully, Skinner et al., (2018) present us with a framework for assessing LEM model performance and this paper uses this methodology to comprehensively assess the relative sensitivity of grid-based LEMs to resolution of the grid cells used, in comparison to changes to important parameter values. It can help us determine whether there is a level of grid independence in LEMs. Importantly, the Morris Method framework also allows us to look at the non-linear influences of changing gird cell size on other key model parameters and on overall model behaviour and efficiency. Figure 1 shows how the Morris Method

ranks parameters in order of relative influence on model behaviours, where higher values of the mean indicate greater relative influence, and higher values of standard deviation indicating greater dependency on the values of other parameters, i.e., non-linearity (Skinner et al., 2018).





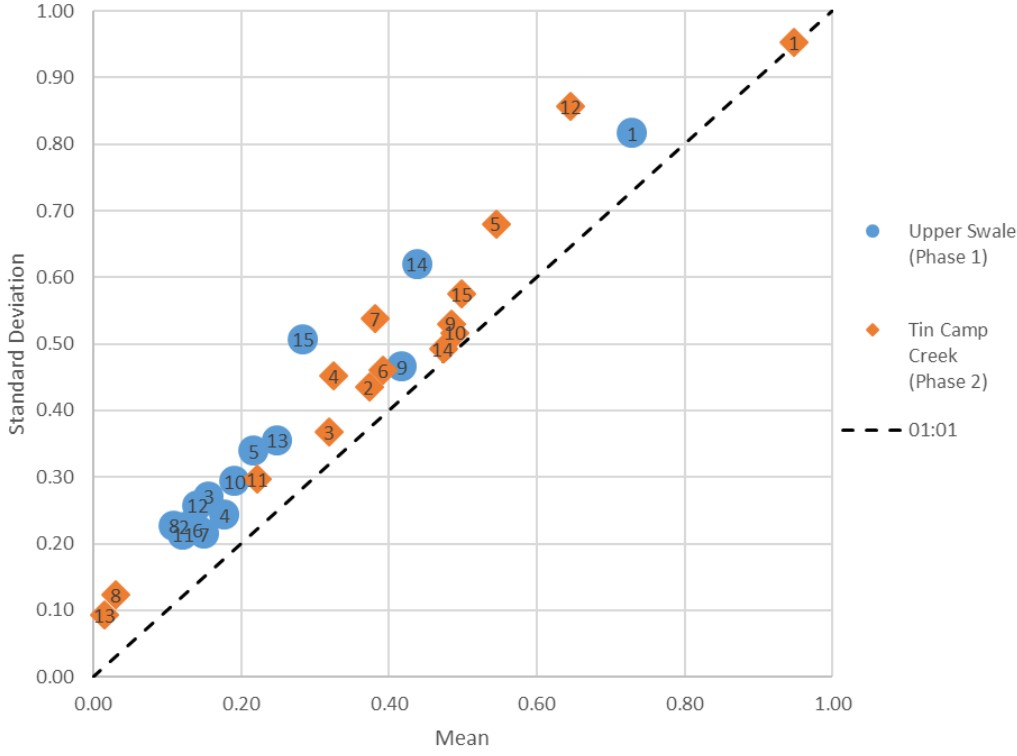


**Figure 1 – : Example of the Morris Method outputs for assessing the sensitivity of the CAESAR-Lisflood model to 15 different model parameters – over two different sized catchments after Skinner et al., (2018). This shows aggregated scores for all Elementary Effects where: 1 = sediment transport formula (SED); 2 = maximum erode limit (MEL); 3 = in channel lateral erosion rate (CLR); 4 = lateral erosion rate (LAT); 5 = critical vegetation shear stress (VEG); 6 =**
**grass maturity rate (MAT); 7 = soil creep rate (SCR); 8 = slope failure threshold (SFT); 9 = in/out difference (IOD); 10 = minimum Q value (MinQ); 11 = maximum Q value (MaxQ); 12 = slope for edge cells (SEC); 13 = evaporation rate (EVR); 14 = Manning's n roughness coefficient (MNR); and 15 = grain size set (GSS).**

Whilst there are few studies addressing DEM resolution on LEM performance, it is clear from the literature that small changes in the landscape (as represented by DEMs) can have an impact on LEM outputs. Therefore, as the spatial resolution of a DEM
affects the representation of topographic features, resolution *will* have an impact on model performance and output. LEMs may be especially sensitive to this as they typically use local gradients to determine erosion and deposition. There are existing methods and frameworks for assessing hydrological and hydraulic model performance and grid independence, however, the chaotic and non-linear behaviour of LEM erosion and deposition patterns may make these methods unsuitable for LEMs. Prior to the use of the Morris Method we have no structure for assessing LEM sensitivity to DEM resolution – or importantly how
model resolution affects the parameter sensitivity.

In this paper we address these issues above and, by using the CAESAR-Lisflood LEM to simulate erosion and deposition over a wide range of spatial DEM resolutions. Outputs from these simulations representing geomorphic, hydrological and model





performance are then assessed using the Morris Method to establish how DEM resolution affects model performance and

results and importantly whether there are any parameter sensitivities to these resolutions.

## 2 Methods

### 2.1 Study Catchment and DEM data

Testing of DEM resolution is carried out on a DEM of Tin Camp Creek in the Northern Territory, Australia (see Figure 2). Tin Camp Creek has a catchment area of $0.5km^2$ and is located within a tropical climate, where its watercourse is ephemeral

– rainfall in the wet season features small, intense convective events. It is a small sub-catchment of the wider Tin Camp Creek system and has been used previously for studies using LEMs (Hancock et al, 2010; Hancock, 2006; Hancock, 2012; Skinner et al, 2018). The DEM used is produced from high resolution digital photogrammetry and available at 2m grid resolution at its finest (as described by Hancock, 2012). For this study the 2m DEM was resampled using the Raster Resample tool in ArcMap v10.4.1 to grid cells sizes between 2 and 30m at 2m iterations and a final 50m grid resolution (see Appendix A). Choice of

small area DEM is deliberate to reduce model run times – especially when using the smallest grid cell sizes. The 2m resolution proved to be too computationally expensive so was not used.

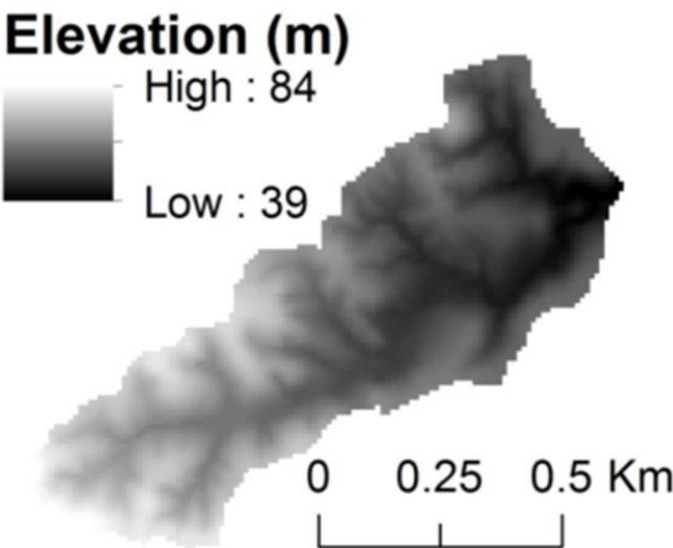

**Figure 2 – Digital Elevation Model of the Tin Camp Creek catchment shown at 10m resolution.**

**2.2 the CAESAR-Lisflood model**

The LEM used is the CAESAR-Lisflood model (Coulthard et al., 2013). A full description of the CAESAR-Lisflood model can be found in Coulthard et al. (2013), and its core functionality is only summarised here. The model utilises an initial DEM



built from a regular grid of cells, and in the catchment mode (as used in this model set up) is driven by a rainfall timeseries that can be lumped or spatially distributed (Coulthard and Skinner, 2016b). At each timestep the rainfall input is converted to
surface runoff using TOPMODEL (Beven and Kirkby, 1979), and distributed across the catchment and routed using the LISFLOOD-FP component (Bates et al., 2010). The CAESAR component of the model drives the landscape development using sediment transport formulae based on flow depths and velocities derived from the LISFLOOD-FP component. Bed load is distributed to neighbouring cells proportionally based on relative bed elevations. This study has not used the suspended sediment processes in the model. The model can handle nine different grain sizes, and information is stored in surface and sub-
surface layers where only the top surface layer is 'active' for erosion and deposition. A comprehensive description of this process can be found in (Van De Wiel et al., 2007).

CAESAR-Lisflood is freely available and since 1996 there have been 119 published studies using the model over a wide range of temporal and spatial scales (Skinner and Coulthard, 2022). Here we used CAESAR-Lisflood v1.9 with modifications to
allow it to run in batch mode and to automatically collect information relevant to the behavioural functions (outlined below).

**2.3 Morris Method**

Our study used the Morris Method (MM) described in Ziliani et al. (2013), i.e. the original MM of Morris (1991), as extended by Campolongo et al. (2007), and applied the "sensitivity" package in the R Statistical Environment (Pujol, 2009) to generate the parameter sets for the Sensitivity Analysis (SA). To set up the MM we selected a number of parameters to be assessed,
specifying a minimum and maximum range for each, plus a number of iterative steps. The parameter values are equally spaced based on the range and number of steps – for example, a parameter with a range of 2 to 10 and 5 incremental steps would have available values of 2, 4, 6, 8, and 10. This was carried out for each parameter and where possible the same number of incremental steps were used for each. For a full description of the MM applied to CAESAR-Lisflood see Skinner et al., (2018) and a summary is provided below.


The MM uses a system of *repeats* to sample the global parameter space. The first test in each repeat uses a randomly selected set of parameters determined from the whole available parameter space. The second test in the repeat uses the same parameter set yet varies a single randomly determined parameter to a different randomly determined value from those available. That parameter would then be excluded from selection for change for the rest of this repeat, with the process continuing until all
parameters have been changed once.

The sensitivity of the model to changes in parameter values is evaluated by the changes of objective function values between sequential tests within repeats, relative to the number of incremental steps the parameter value has been changed by (where for a set of values 2, 4, 6, 8, and 10, a switch from an initial of 4 to either 2 or 6 is one step, a change to 8 is two steps, and 10
is three). The change in objective function score between two sequential tests divided by the number of incremental step



changes is an elementary effect (EE) of that objective function and the parameter changed (Equation 1). After all tests for each grid resolution have been performed, the main effect (ME) for each objective function and parameter is calculated from the mean of the relevant EEs – the higher the ME the greater the model's sensitivity. Alongside the ME, the standard deviation of the EEs is also calculated as this provides an indication of the non-linearity within the model.


**Equation 1**

$$d_{ij} = \left| \frac{y(x_1 x_2 \dots, x_{i-1}, x_i + \Delta_i, x_{i+1}, \dots, x_k) - y(x_1 x_2 \dots, x_{i-1}, x_i, x_{i+1}, \dots, x_k)}{\Delta_i} \right|$$

where $d_{ij}$ is the value of the $j$th EE ($j = 1, \dots, r$; where $r$ is the number of repetitions (here $r = 100$)) of the $i$th parameter (e.g.

$i = 1$ refers to sediment transport formula, see Table 1), $x_i$ is the value of the $i^{th}$ parameter, $k$ is the number of parameters investigated (here 7), $y(x_1, x_2, \dots, x_k)$ is the value of the selected objective function, and $\Delta_i$ is the change in incremental steps parameter $i$ was altered by.

In Skinner et al., (2018) a MM test was applied to a DEM of the same Tin Camp Creek basin used for this study, using a single 10m grid cell size (see Figure 2). That test used a sub-set of 15 parameters and 100 repeats, producing 1600 tests in the total. Here we are testing 15 different resolutions so the same level of scrutiny of parameters and repeats would result in 24,000 tests to be required. To reduce the computational expense, we reduced the 15 parameters to the 7 that exhibited the greatest impacts on model behavior in Skinner et al., (2018). In addition, the number of repeats was reduced to 10 in line with the minimum

number suggested by Ziliani et al., (2013). This reduced the total number of tests to 80 for each DEM resolution and 1,200 in total. The only difference in the parameter value ranges to Skinner et al., (2018) is the inclusion of Meyer-Peter Muller (Meyer-Peter and Muller, 1948) as an additional sediment transport law, with changes between any of the sediment transport laws being counted as a single iterative step change. Meyer-Peter Muller was included as this formula had been added to version 1.9d of CAESAR-Lisflood since the Skinner et al., (2018) analysis and enabled a greater number of 'steps' to be included in

the sediment transport component. The parameters and their values are shown in Table 1.

**Table 1 – Parameters selected for the MM test, the number of iterative steps applied, and the values used for each iterative step.**

| Code | Parameter | Steps | Tin Camp Creek |
|------|-----------|-------|----------------|
| (1) SED | Sediment Transport Formula | 3 | 1 Wilcock & Crowe / 2 Einstein / 3 Meyer Peter Muller |
| (2) GSS | Grain Size Set | 5 | Set 1; Set 2; Set 3; Set 4; Set 5 |
| (3) MNR | Manning's n Roughness | 5 | 0.03; 0.0325; 0.035; 0.0375; 0.04 |





| (4) TOPN | m value used by TOPMODEL | 5 | 0.005; 0.0075; 0.01; 0.0125; 0.015 |
| (5) VEG | Vegetation Critical Shear Stress (Pa) | 5 | 2; 3.25; 4.5; 5.75; 7 |
| (6) MAT | Grass Maturity Rate (yr) | 5 | 0.5; 0.875; 1.25; 1.625; 2 |
| (7) MEL | Max Erode Limit (m) | 5 | 0.001; 0.0015; 0.002; 0.0025; 0.003 |


In Skinner et al., (2018) a model function approach to evaluating MM was developed and tested using the CAESAR-Lisflood model. The main purpose of the model function approach was to mitigate for the fact that there is almost always a lack of suitable observation data to use in evaluating the performance of LEMs via an objective function approach (an objective function being the error score between modelled and observed data). Instead, Skinner et al., (2018) proposed a series of metrics that would assess key behaviours in the model relating to its outputs and assess MM against changes in the model's behaviour. Here we use the same 15 model functions as Skinner et al (2018) and these are shown in Table 2. To summarise the large amount of information produced, the ME of each parameter and model function combination was normalised based on the proportion of the ME for highest ranking parameter for that model function – therefore the highest ranked parameter for each model function always scored 1. The scores for each parameter were aggregated for across all model functions based on the mean of the scores. The model functions were further sub-divided into core behaviour groups (Table 2) and the scores aggregated again for each core behaviour. The same was also done, separately, for the standard deviations of each parameter and model function.

**Table 2 – Core behaviours of the model and the Model Functions adopted to assess changes to these (from Skinner et al, 2018).**

| Core Behaviour | Model Function |
| --- | --- |
| Catchment Sediment Yield | Total Sediment Yield |
| | Mean Daily Sediment Yield |
| | Peak Daily Sediment Yield |
| | Time to Peak Sediment Yield |
| | Days when Sediment Yield > baseline |
| Internal Geomorphology | Total Net Erosion |
| | Total Net Deposition |
| | Area with > 0.02m Erosion |
| | Area with > 0.02m Deposition |
| Catchment Hydrology | Total Discharge |
| | Mean Daily Discharge |





| Model Efficiency | Peak Daily Discharge |
|---|---|
| | Time to Peak Discharge |
| | Days when Discharge > baseline |
| Model Efficiency | Total Model Iterations |

The same set of repeats and parameter changes were used for each grid cell size to allow direct comparison. Finally, using the
full set of results across all of the grid cell sizes a further analysis was performed with grid cells size as an additional parameter
to assess its relative influence on the model compared to the other 7 parameters. This was done by using 5 steps (4, 8, 12, 16,
and 20m resolution) and randomly selecting the starting grid cell size for each repeat, the position in the sequence it is changed,
and the change in steps.

Each individual test within the repeats consisted of 30 years of simulations using the same input rainfall. The rainfall was
produced by using 23 year observation record from a single raingauge at Jabiru Airport, with the first 7 years repeated for the
full 30 year input. This was applied as a lumped input at a 1h timestep.

## 2.4 Stream network analysis

To examine how stream network metrics changed with DEM resolution, the Hydrology tools in ArcMap 10.4.1 were used to
extract stream networks and stream orders for each resolution with the Strahler (1957) and Shreve (1966) methods. Both
methods are top-down approaches to stream ordering in that they start from the source and increase in value towards the outlet.
The Strahler method (Strahler, 1957) calculates the depth of the drainage network – a 2nd order stream begins only where two
or more 1st order streams meet, and a 3rd order only where two or more 2nd order stream meet, and so on. As such the maximum
stream order number does not provide information on the number of individual streams within the network. The Shreve method
(Shreve, 1966) assigns stream values cumulatively, so where two streams with a value one meet, the downstream becomes 2,
and unlike the Strahler method, lower order tributaries are including in the ordering, so where a stream with value 1 joins with
a stream with value 2, the downstream section is assigned a value of 3. Therefore, the stream number at the outlet provides
information on the number of streams in the system. In ArcMap a stream identification threshold (in effect delineating 1st order
streams) was set at 1000m$^2$ - rounded up to the nearest whole pixels. The total number of 1st order streams calculated by the
Strahler method was estimated by converting the raster output to a polyline and selecting only 1st order streams. With the
exception of the 2m resolution, the analysis was performed on the post-test DEM for Test 1 of each resolution test to allow a
drainage network to be established on spun-up into the DEM surface – this negated the need to pit fill the DEM before the
network analysis.





## 3. Results

### 3.1 Influence of grid cell size on model outputs (Model Functions)


The ME for each of the 15 model functions were calculated for each of the grid cells sizes and the patterns observed for each Model Function are summarised in Table 3. The box and whisker plots of Figure 3 highlights these patterns for four of the Model functions (plots for all of the Model Functions can be viewed in the Appendix B). Figure 3 shows that the mean total erosion and sediment yields output by the model remain similar for grid resolutions up to 24-30m, although the spread of

values across the 80 tests vary more with larger grid cells. However, the peak daily sediment yields increase with larger grid cells, whereas there is a decrease in the number of days where sediment yield is over the baseline. This indicates that with larger grid cells there are less events that produce erosion and sediment outputs offset by an increase in erosion and sediment outputs during larger events.

**Table 3 – Visual interpretation of influence of grid resolution on the 15 model functions.**

| Model Function | Pattern with Coarse Grid Resolution | Resolution where Patterns breaks down |
|---|---|---|
| Total sediment yield | little change | >22m |
| Mean daily sediment yield | little change | >22m |
| Peak daily sediment yield | increase | >24m |
| Time to peak sediment yield | little change | >26m |
| Days when peak sediment yield > baseline | decrease | - |
| Total net erosion | little change | >22m |
| Total net deposition | decrease | >10m |
| Area with > 0.02m erosion | decrease | - |
| Area with > 0.02m deposition | little change | - |
| Total discharge | decrease | - |
| Mean daily discharge | decrease | - |
| Peak daily discharge | little change | >20m |
| Time to peak discharge | increase | >20m |
| Days when discharge > baseline | decrease | - |
| Total model iterations | decrease | >12m |






**Figure 3 – Box and whisker plots showing the mean and spread of model outputs at each grid resolution, for (a) Total Sediment Yield, (b) Total Net Erosion), (c) Peak Daily Sediment Yield, and (d) Days over Sediment Yield Threshold.**


The box and whisker lots in Figure 3 aggregate the results of 80 tests and could conceal a range of varying model behaviours when using the same parameter set across the different grid cell sizes. To check this, 5 parameter sets, or tests (numbered as the order they appear in the MM), were randomly selected and plotted in Figure 4. This shows that the individual tests follow the overall trends displayed in Figure 3.






Earth **Surface**
**Dynamics** Open Access
Discussions

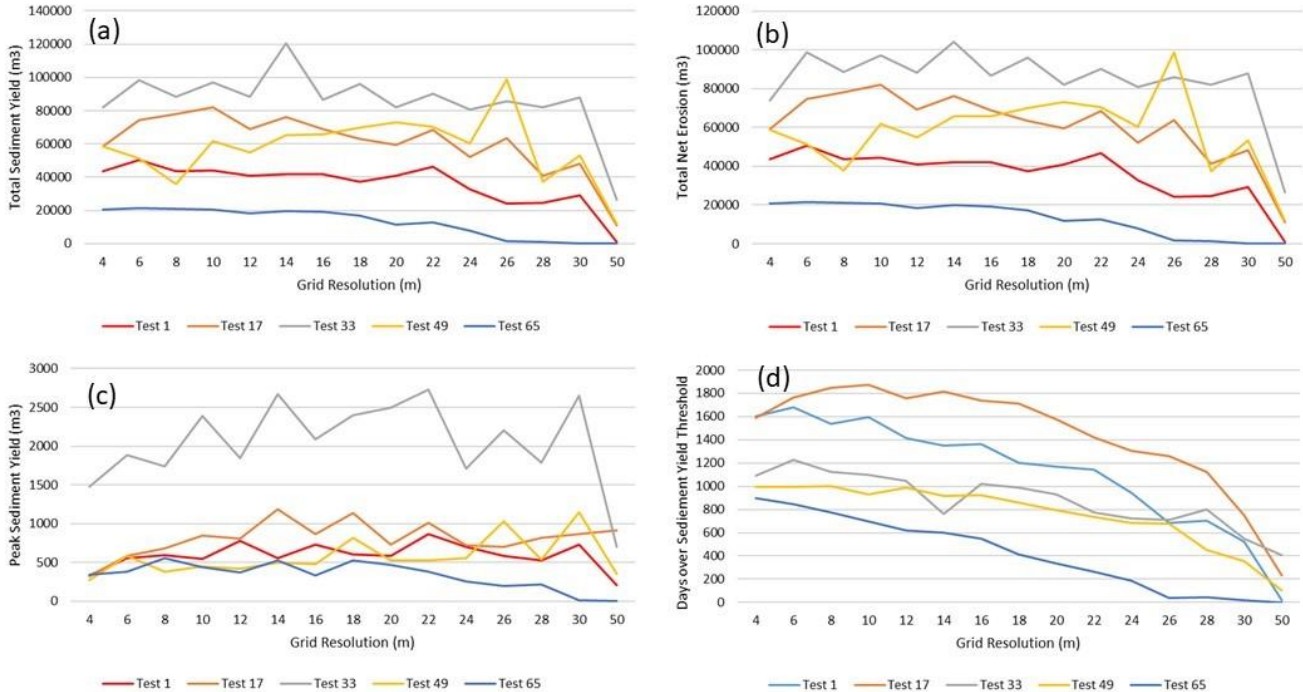

**Figure 4 – Output values for five randomly selected test numbers at each grid resolution, for (a) Total Sediment Yield, (b) Total Net Erosion, (c) Peak Sediment Yield, and (d) Days over Sediment Yield Threshold.**

### 305  3.2 Influence of grid cell size on model behaviour (summary of MM)

The mean aggregated ME scores, measuring the overall relative influence of each parameter across all of the model functions, are shown in Figure 5. Although there is variation between the grid cells sizes, the relative influence of each parameter remains fairly consistent across DEM resolution. The clear exception here being the Sediment Transport Law, that whilst remaining the most influential parameter for the majority of resolution, its relative influence decreases as the grid coarsens, until for the

coarsest of grids the TOPMODEL M replaces it as most influential (at 28m and 50m). It appears that it is an increase in influence of the TOPMODEL M that drives the decrease below 26m, beyond which all the other parameters increase in influence as Sediment Transport Law further decreases.





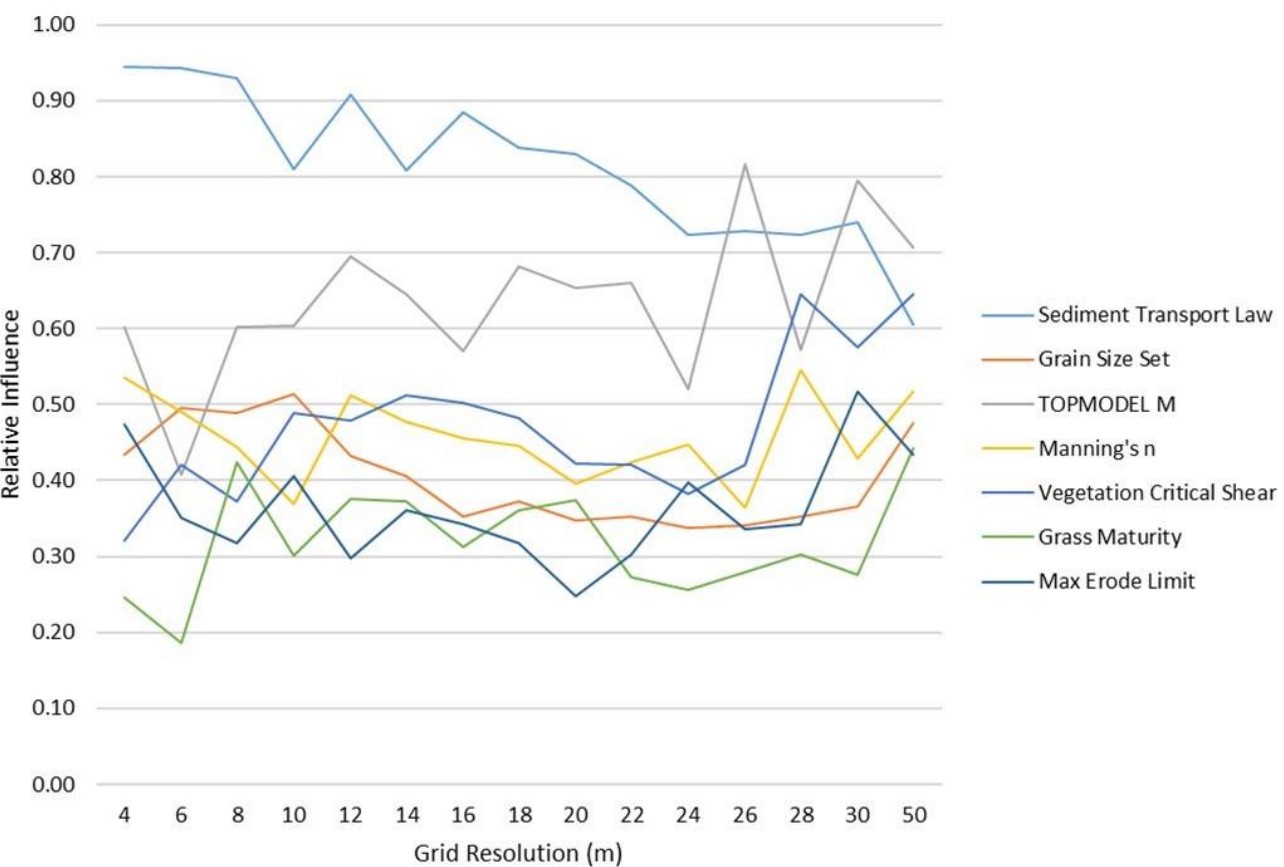

**Figure 5 - Mean ME scores aggregated for all Model Functions for each parameter and each grid cell size.**

### 3.3 Stream network analysis

The numbers of stream orders calculated is shown in Figure 6. The maximum number of stream orders calculated by the Strahler method remained consistent throughout, dipping to three orders at 28m resolution, and three again at 50m. The Shreve

and Strahler first order counts steadily decreased as grid cells become larger. This shows that the depth of the drainage network does not reduce until the largest grid cells are used. However, the detail within the network is being lost with less 1st Strahler orders, and lower Shreve numbers, with larger grid cells. The disparity between the Shreve number and the number of 1st Strahler order streams is due to disconnection of part of the drainage channel to the main network, therefore these stream not contributing to the Shreve number. This disconnection was not consistent through the resolutions, with some coarser resolution

displaying a better connected network than others.





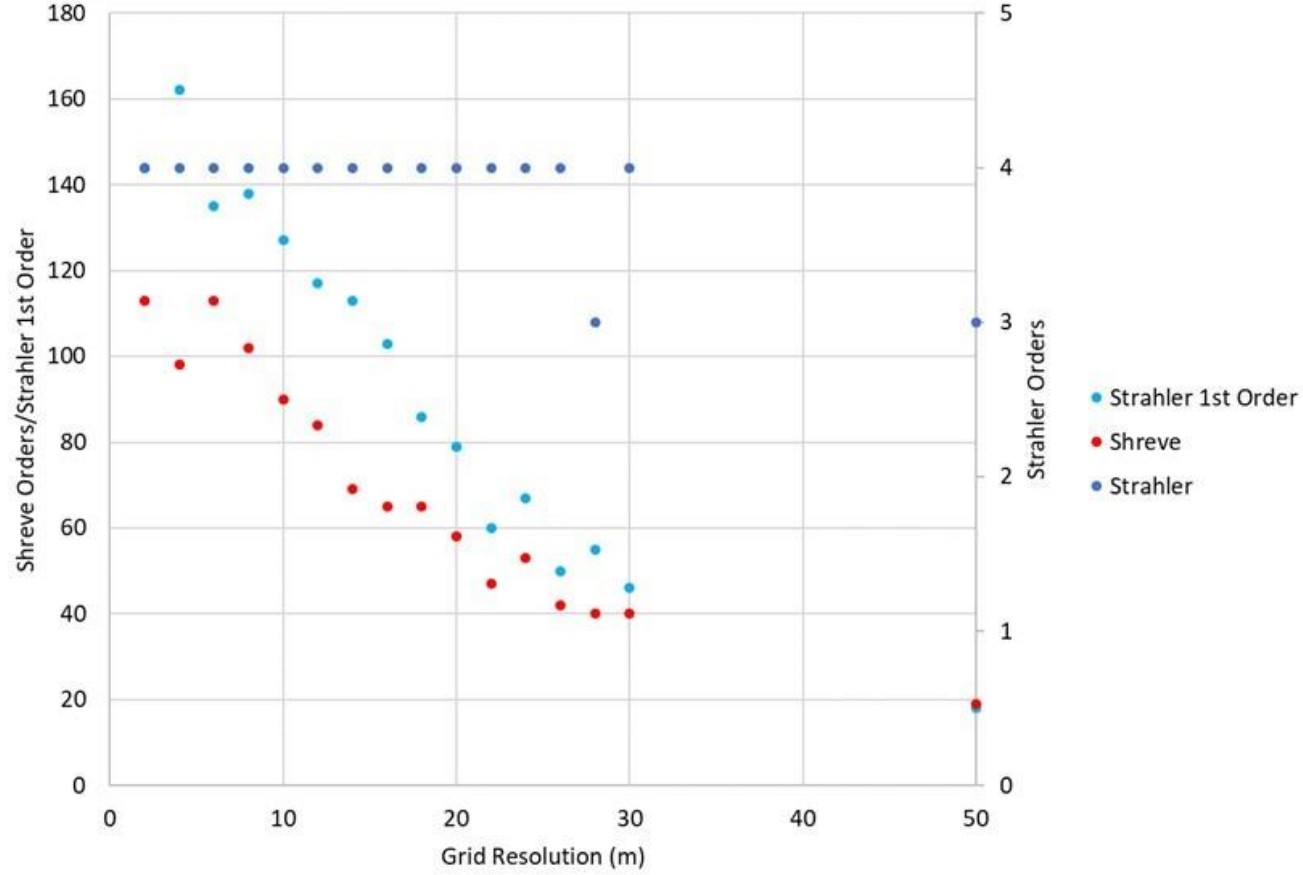

**Figure 6 – The maximum number of stream orders calculated at each grid resolution using both the Strahler and Shreve methods. The total number of 1st Order stream cells calculated by the Strahler method is also shown.**


### 3.4 Model performance

Figure 7 shows the changes in the number of iterations required by the model with each grid cell size. The iterations represent the number of calculations required by a test and is a useful proxy for model efficiency that is independent of the specification and performance of individual machines. There is a visible rapid drop off in the number of iterations required between 4m and

12m resolutions, yet little change with increasing coarseness beyond 12m, suggesting there are only marginal computational efficiency gain to be made using DEM resolutions coarser than 12m (the spread of the total number does decrease beyond 12m still).





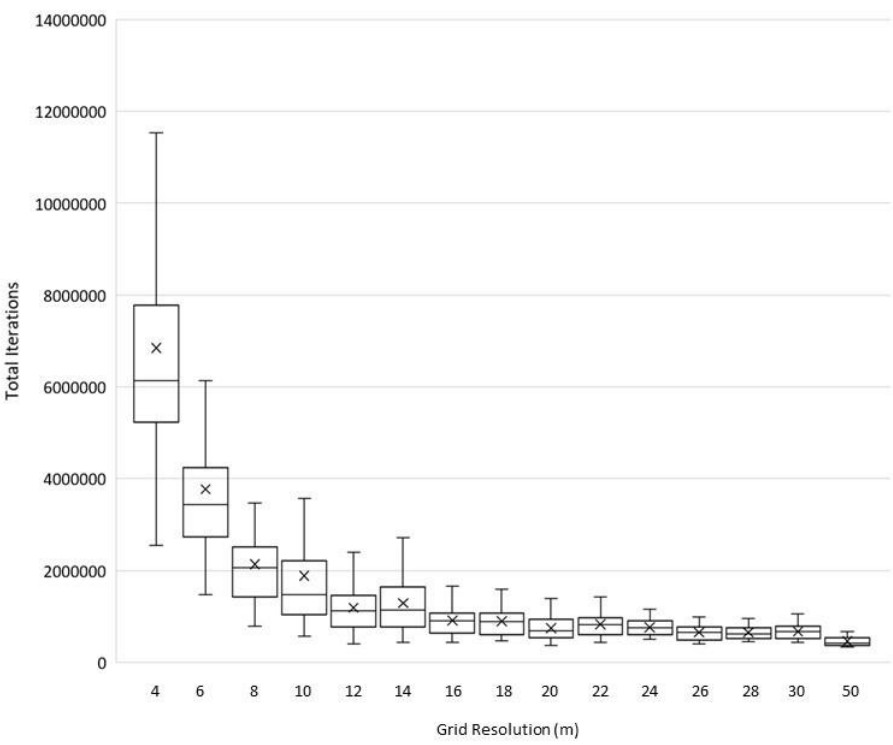

**Figure 7 – Box-Whisker plots showing the spread of total number of model iterations required at each grid resolution.**

### 3.5 Relative influence of grid cell size

The results of from the MM tests for each grid cell size were further used to simulate a MM run where the DEM grid resolution could be considered a parameter itself. Figure 8 summarises the results and reveals the relative importance of grid cells size when compared to the key parameters in the model. Overall, it has a similar level of influence over all core behaviours as the

Sediment Transport Law (i) and the TOPMODEL M. Broken down, this is skewed by the Catchment Hydrology (iii) and Model Efficiency (v) core behaviours where it is the most influential parameter, whilst it has relatively low relative influence on the Catchment Sediment (ii) and Internal Geomorphology (iv) behaviours.





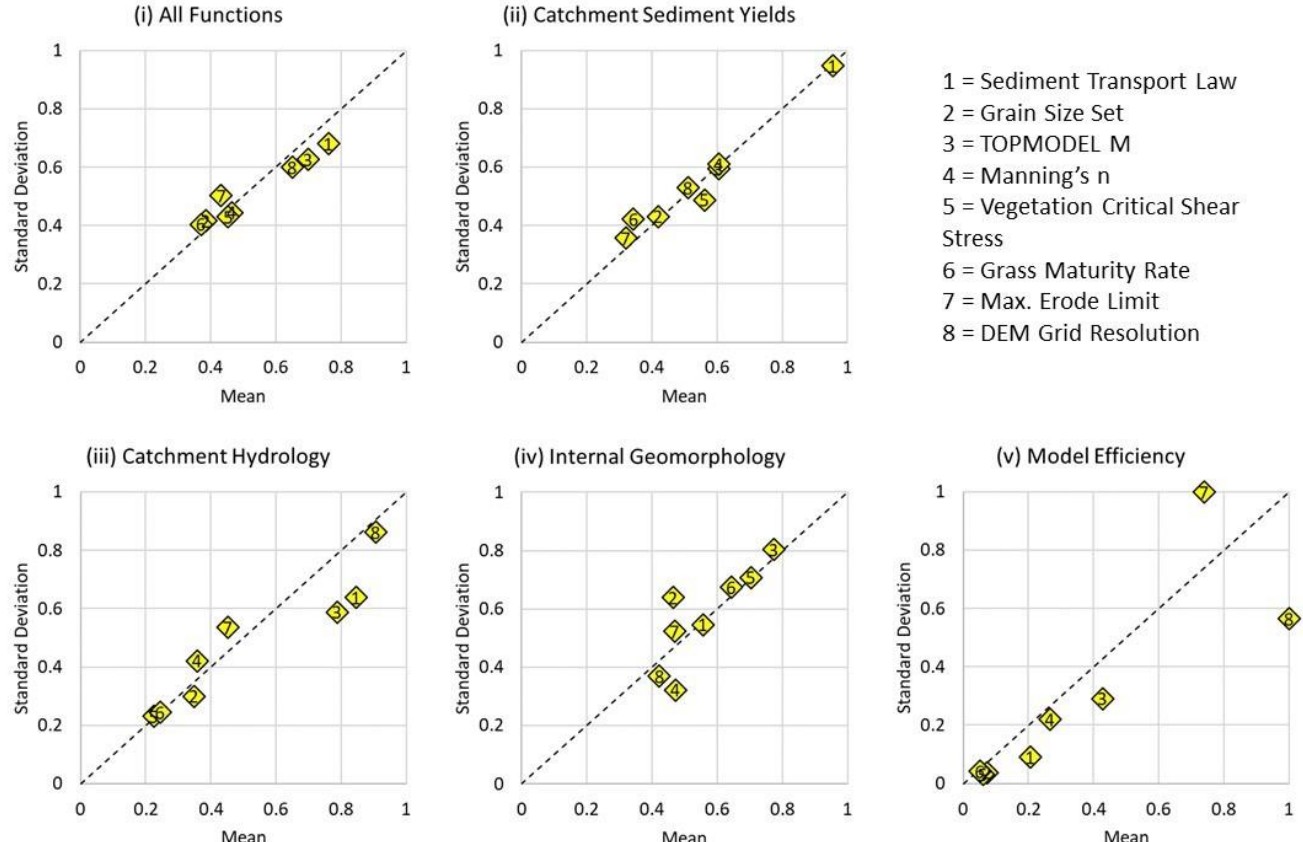

**Figure 8 - Mean and Standard Deviation of relative MEs for the seven parameters and DEM Grid Resolution for all Model Functions aggregated by Core Behaviours (see Table 2).**

## 4. Discussion

**4.1 Model robustness to grid resolution**

Certain aspects of CL's performance are relatively robust to changes in grid cell size. However, this behavior can conceal important differences. As shown in Table 3 and Figures 3 and 4, important factors including Total Sediment Yield and Total Net Erosion display little change until grid cell sizes >22m. However, whilst these output metrics remain relatively constant,

related factors Peak Daily Sediment Yield increase, and Days When Peak Sediment Yield is > Threshold decrease. Despite long term sediment yields being similar this demonstrates a change in model behaviour, where with larger grid cells the sediment delivered from the catchment is doing so in fewer yet larger bursts. This can be explained by a loss of the granularity





of the drainage network as grid cell size increases, as shown in Figure 9. In particular, 1st Order streams are lost with larger grid cells (see Figure 6). With smaller grid cells there is a more detailed channel network, so when summed across the whole

basin the process of channels geomorphically 'switching on and off' is smoother. With larger grid cells, there are less channels meaning the switching on and off response is more step like and thus more spikey. This also illustrates one of the weaknesses and difficulties of using a lumped parameter such as basin sediment yield to describe both model performance and basin geomorphology. There was also a decrease in Total Net Deposition with larger grid cells, consistent with the findings of School et al, (2000) that using larger grid cells makes the conditions needed for sedimentation less likely to occur. We did not see any

of the larger grid  dependent erosion fluctuations demonstrated by Nicholas (2005) with their braided river model, that can be explained by CL using the flow velocity derived from water surface slope to calculate bed shear stress and thus sediment transport rather than bed slope. In other LEM's based on bed slope this sensitivity may therefore remain.

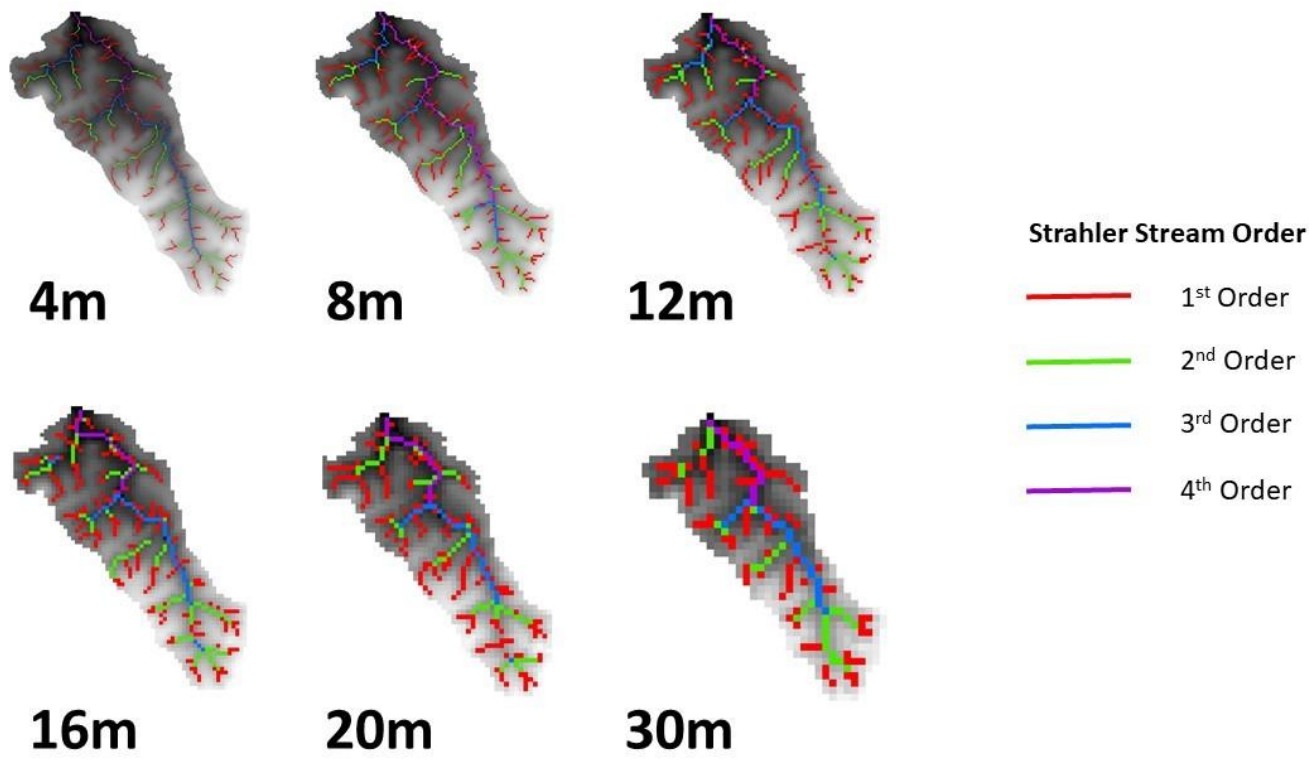

**Figure 9 – Illustration of the loss of granularity in the representation of the stream network (shown is red) with increasingly large grid cell sizes.**

**4.2 Model behaviour with grid resolution**



The behaviours of the model are captured by the relative influence of parameters on the Model Functions and Core Behaviours, with changes in the relative influence being taken as a change in behaviour. The relative influence of the 7 parameters tests on the Core Behaviours when using different grid cell sizes is summarised in Figure 5 and suggests a similar story to the changes in factors that would indicate model robustness. Although there is some noise, the relative importance of the different parameters remains relatively unchanged below the coarsest resolutions > 22m. However, there is also a general trend of

waning influence of the choice of Sediment Transport Law that appears to begin at 10m resolution and continues throughout. This indicates that the coarsening of the grid resolution is resulting in some key behavioural shifts in the simulations that are most likely related to the loss of detail in the drainage network discussed in 4.1.

Relative to the 7 parameters used in the MM tests, grid cell size showed a mixed degree of influence on the model behaviours

(see Figure 8). It had the greatest influence on the Model Efficiency, which also had a relatively low standard deviation ME suggesting this was non-linear and other parameter values did not affect this. This is exactly as would be expected as halving the grid cell size increases cell number by four, which also increases the number of iterations required by a same degree. Grid cell size had the greatest influence over the Catchment Hydrology behaviours, which is consistent with the loss of detail in the drainage network discussed in 4.1, and also the findings of Savage et al (2016) that grid resolution influences the sensitivity

of hydraulic models to parameter values. It had relatively less influence over the Catchment Sediment and Internal Geomorphology behaviours but still influential enough to require consideration. Overall, grid resolution was shown to be the third most influential parameter, to a similar degree as the Sediment Transport Law and TOPMODEL M.

### 4.4 Implications and suggestions


This work has only simulated the influence of grid cell size on a single LEM and on a single, relatively small, catchment. The role of the loss of detail in the drainage network with larger grid cells is a physical effect applicable across all models using a regular grid of elevation and to any catchment, regardless of size or situation. However, the relative impact will change with different size basins – at different resolutions – and quite possibly when experiencing a different range or magnitude of driving

events. Therefore, the generic finding of our study – that the degradation of a model DEM (larger grid cells) conceals topographic details that may be important to model outcomes – is important. But, the actual impact on individual studies will be specific to the catchment modelled and the resolution chosen.

This study has further highlighted the need for operators to better understand the sensitivities of their models to both internal

and external factors before embarking on landscape evolution studies. Here, we have demonstrated that DEM grid resolution is a controlling external factor of the behaviours simulated in the system, with larger grid cells resulting in fewer yet more extreme erosion producing events, which although overall produces similar total sediment yields does so over a smaller

Earth **Surface**
**Dynamics**
Discussions

contributing area. Grid cell size was also shown to be the third most influential parameter out of those tested. This suggests that it is important that when making model choices, operators should aim to use the finest resolution available to them and

that model efficiency will allow. Where model efficiency is a concern, a compromise can be made by selecting a resolution where coarsening further results in lower levels of benefits - in this study that would appear to be around 12m (Figure 7), close to the default 10m used for the same catchment in Skinner et al., (2018). Importantly – this choice is catchment specific.

The use of MM to assess the model's sensitivity to an external factors (grid cell size) relative to the sensitivity to internal

parameters presents opportunities and a powerful tool. This allows for a more comprehensive consideration of the sensitivity of the model to choices of the modeller and enable them to evaluate on what areas to focus on improving, for example, the tests here tell the modeller that work to increase the resolution and detail of the DEM grid would yield greater benefits than work to reduce the uncertainty in the grain size parameters. Performing this type of analysis to inform model choices will increase the robustness and confidence in model outputs, crucial if LEMs are going to be used operationally and for decision

making. Additionally, the same methods can be applied to other modelling fields, including hydrological and hydraulic modelling.

## 5. Conclusions

This research has explored the influence DEM grid cell size has on the water and sediment outputs of a LEM (CAESAR-Lisflood) using the Morris Method sensitivity. We found simulated basin sediment yields and hydrological outputs totalled over the model duration, were largely unchanged as grid cells sizes increased, up to a point where the grid cell size started to degrade the extent and shape of the drainage network. It is likely that the impact of this network degradation will be dependent on the size of the basin, with the results being lessened on larger basins for the same grid cell size.


However, when the model results are analysed over event scale timescales it became clear that the lumped output grid scale independence masked important changes with resolution. As grid cell sizes increased, the similar sediment yields were produced by fewer, larger events. It is important, therefore, to note this sensitivity in the models application or risk a 'right results for the wrong reasons' set of outputs.


These findings are important because the resolution of the DEM used in LEM studies is often an arbitrary choice, often driven by the need to balance including as much detail as is available with model efficiency. The approach presented in this paper demonstrates the feasibility of using a screening sensitivity analyses to identify key influences on model behaviour, for grid cell size and parameter choices, which will help modellers identify the optimal grid cells size for their study.




## Appendix A - DEMs at different grid resolutions

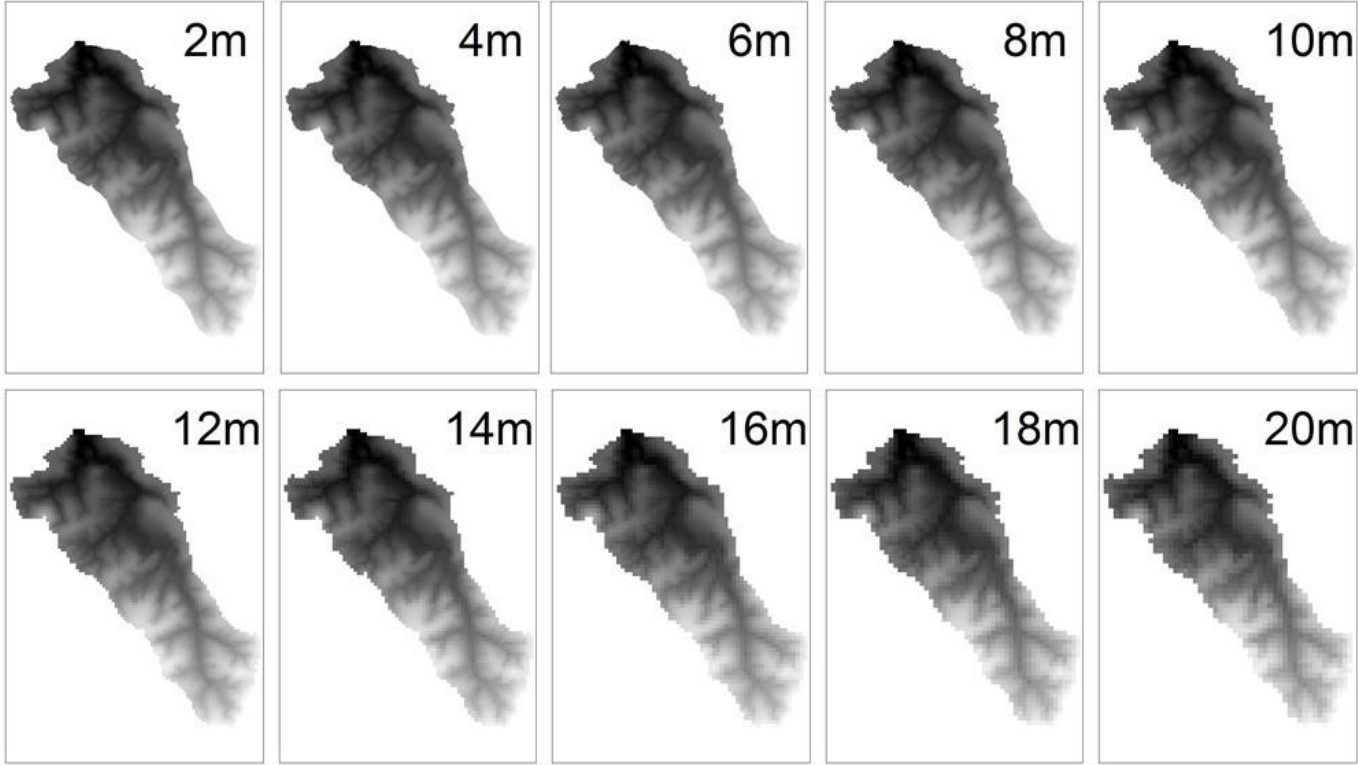



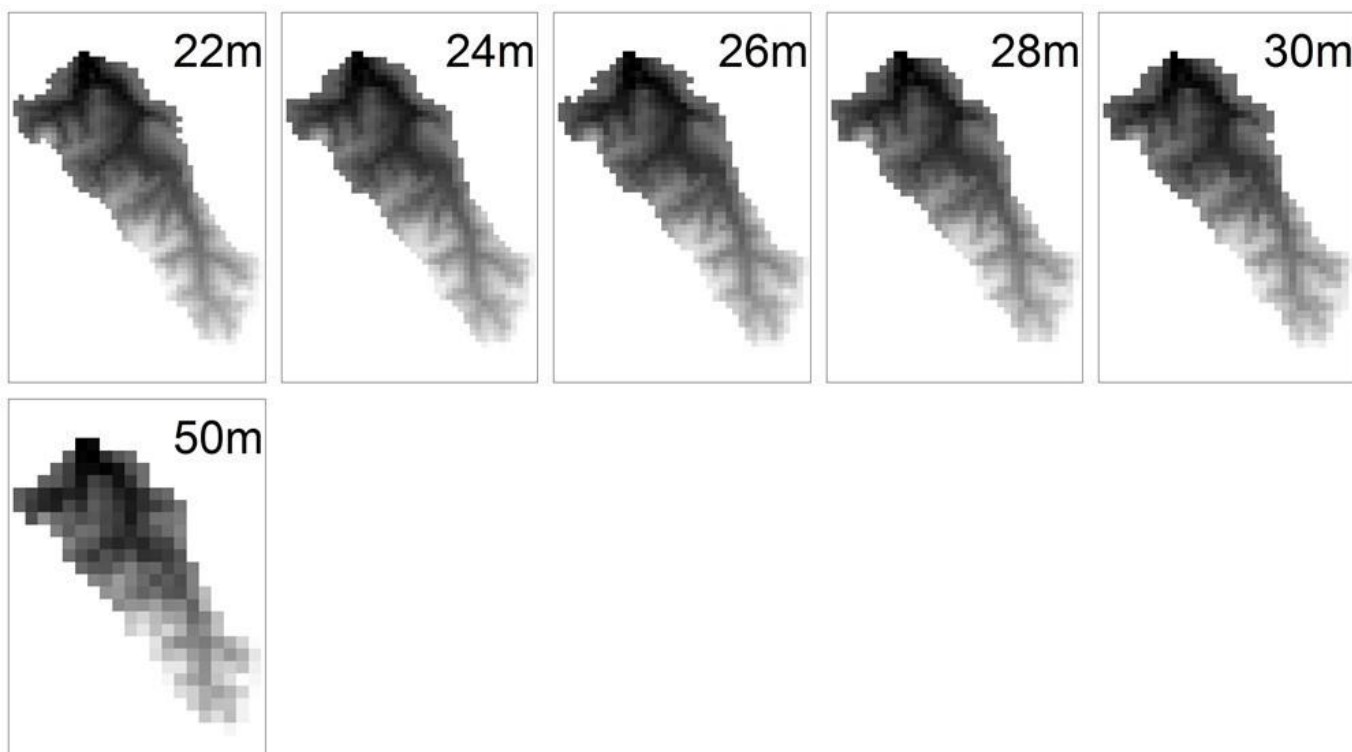

**Figure A1 - Illustrations of the Tin Camp Creek DEM resampled at 2m increments, from 2m grid resolution to 30m, plus 50m.**



## Appendix B - Box and whisker plots



**Figure B1 - Box and whisker plots showing the spread of model outputs, and the means at each grid resolution.**

## Acknowledgements

This work was funded by the NERC Flash Flooding from Intense Rainfall (FFIR) project, Susceptibility of Basins to Intense Rainfall and Flooding (SINATRA) NE/K008668/1. The CAESAR-Lisflood code is freely available from https://sourceforge.net/projects/caesar-lisflood/ or http://www.coulthard.org.uk





## Author Contributions

CS and TC conceived and designed the project. CS set up the tests and performed the model runs. CS and TC analysed the results, wrote, and edited the manuscripts.

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
