# Peer review of "Testing the sensitivity of the CAESAR-Lisflood Landscape Evolution Model to grid cell size"

_Earth Surface Dynamics, 2022_

## Referee Comment (RC2)

[referee-annotated manuscript omitted]

---

## Author Comment (AC1)

**Dear Editor**

We would like to thank both reviewers, John Armitage and anonymous Reviewer #2, for their insightful and useful comments. Below we detail how we have revised the manuscript according to the reviewers comments and suggestions.

**Reviewer #1**

In this manuscript the authors discuss the sensitivity of the LEM CAESAR-Lisflood to grid cell size. They find that for a series of model functions that the numerical model is roughly as sensitive to DEM resolution as two other important parameters, the sediment transport law, and the TOPMODEL m parameter (that controls the peak and recession curve for the transformation of rainfall to runoff). There is the important observation that despite resolution having some, but perhaps not a significant impact on model functions such as total sediment yield or time to peak sediment yield, there is the potential that the model gives the "right answer for the wrong reasons".

In this manuscript the authors re-use the experimental method developed in Skinner et al. (2018) to use the Morris Method to explore the relative importance of parameters to one and other. This is achieved (please correct me if I am wrong) by first selecting a representative sample of models to run (1220 models in this case) and then plotting the standard deviation and mean effect of each parameter relative to a model function (Table 2 in the manuscript). From this the most important factors might become apparent. From reading up on the subject, I see that the Sobol method however gives a quantified effect of each parameter on the model result, however in Skinner et al. (2018) it is stated that the Morris Method is good enough given evidence from previous studies. It would be interesting to see this point demonstrated, but perhaps that is a technical point for some future study.

The Morris and Sobol methods are different in purpose. The Sobol method is a full global sensitivity analysis and thus provides a quantified effect as stated. The Morris method only samples the global space and is meant to provide a less computationally and time expensive screening to identify the most influential parameters before assessing them with a more comprehensive method like Sobol. Our intention here is to demonstrate how Morris can be used as a relatively efficient method to help guide modellers in setting up CAESAR-Lisflood rather than performing full sensitivity analyses on the model. We do agree, this would make an interesting useful future study.

I think this manuscript is a useful contribution to understanding the operational use of landscape evolution models that are process based, such as CAESAR-Lisfood. From using this code, my colleagues and I have noticed that for example the Mannings coefficient needs to be adjusted if a higher resolution DEM is used to replace a coarse resolution DEM. This manuscript starts to put these sorts of "tunings" into context of the limitations of the model approach.

This is an interesting insight and would be useful to other users of CAESAR-Lisflood.

*I have a few comments that the authors might find useful to further improve this manuscript (in no specific order):*

It would be useful to see to what extent the change in DEM resolution impacts the spatial distribution of erosion and deposition. There is a focus on the model functions in Table 2, for obvious reasons, however these are spatially lumped. The biggest advantage of using a code like CAESAR-Lisflood is it can be used to model the spatial distribution of erosion and deposition. If I were interested in only gauging station measurements of water flux and sediment yield, I could turn to one of the many 1D models that treat the river network as a line and get over the problem of resolution. Therefore, it would be ideal to get some feeling for how resolution impacts the spatial distribution of landscape change. If the authors think it possible, perhaps some analysis of the DEMs of difference between the start and of each model could be analysed. A plot of the distribution of the mean and standard deviation of elevation change for each DEM resolution? Or the same exercise for each sub-catchment in the DEM as a function of model resolution? It would be interesting to see at what resolution the spatial distribution of erosion and deposition starts to converge.

We wanted to do this – but struggled to find a way of describing this other than qualitatively. Some of the model functions, including area subject to erosion and area subject to deposition picks up some of what is happening internally, but obviously not in great detail. The means and SD's of the elevation changes tend to be heavily influenced by cells that erode/deposit a small amount (with only a few showing much erosion/deposition) and we struggled to make such metrics work – especially having to consider normalising/accounting for the different numbers of grid cells involved in the different resolutions. We could spend more time on this but would suggest that at present it is outside our scope but certainly an interesting area of future work.

The sensitivity analysis is carried out on an existing landscape, where the landscape features already exist. Another application of LEMs is to try and model landscape formation. Here the impact of model resolution might be more acute, as the channels have not been carved into the landscape, and the model equations are free to form the landscape features. It would be interesting to run the same sensitivity analysis on a simple slope, perhaps with some noise to localize the flow routing algorithm. This would confirm the robustness of the results from the Morris Method that suggest DEM resolution is as important as the sediment transport law and the TOPMODEL m parameter. Spatial statistics, such as the wavelength of valley spacing, could also be measured to discover if below a certain resolution the model reproduces the same topography (e.g. Armitage, ESurf, 2019; https://doi.org/10.5194/esurf-7-67-2019).

We agree this would be a really interesting experiment to perform but outside of the scope of this work. The motivation of the work was to consider the sensitivities in a context analogous to uses of the model that support real-world decision making, rather than the more traditional experimental function of LEMs.

Why was the Meyer-Peter Muller sediment transport model introduced in this study? How has it been included? What are the benifits of using it over Wilcox and Crowe? What are the drawbacks?

Simply because MPM has been added CAESAR-Lisflood since the previous study and this functionality allowed us to test three sediment transport rules as opposed to two. An advantage is that we demonstrate how the method can be used to help guide user choices in model set up with regard to more choice in the sediment transport rule. In case it's not clear, MPM was included in addition to Wilcox and Crowe, not in place of.

The sediment transport rules present landscape modellers with a real issue – in that we demonstrate that this one largely subjective choice has the greatest impact on model outcomes. The implication is that we cannot state whether there is a right answer without some form of calibration exercise. Sed TPT rules are a known weakness / knowledge gap in fluvial geomorphology and we think this paper reflects that rather than seeking to comment on it. All are wrong, but some are useful etc.

I recently read the chapter "Transport of gravel and sediment mixtures", in Sedimentation Engineering: Theories, Measurements, Modeling, and Practice, by Gary Parker (2008; https://ascelibrary.org/doi/10.1061/9780784408148.ch03). In this chapter he states that "Einstein (1950) was the first to execute such an analysis for the bed-load transport of mixtures. The relation cannot be considered appropriate for the purposes of calculation due to the gross inaccuracies in the hiding function." I am curious as to why the Einstein model remains within this analysis if it is known to badly represent the hiding effect of large grains on smaller grains?

Einstein, MPM and W&C are used in CL because they all calculate the total sediment transported as a sum of the amount moved for each grainsize category. This fits well with how CL is set up. These formulations are largely chosen for these operational/coding purposes rather than one being better or worse than the other. Though – by having W&C and Einstein we have two sediment transport rules that include and don't include hiding effects. Making them useful for different fluvial environments.

**In Figure 5, what is the "vegetation critical shear" and the "grass maturity"?**

These are parameters relating to the vegetation model in CAESAR-Lisflood and are described in Skinner et al (2018). Vegetation Critical Shear is the shear stress threshold that strips vegetation in pixels, whilst Grass Maturity is the rate it takes vegetation in pixels to grow to full maturity. We have added a table in Appendix A that describes the roles these parameters play in the model -

| Code     | Parameter                             | Description                                                                  |
|----------|---------------------------------------|------------------------------------------------------------------------------|
| (1) SED  | Sediment Transport Formula            | Relationship between properties of flow and the volume of sediment           |
|          |                                       | entrained, often determined via field and laboratory observations.           |
| (2) GSS  | Grain Size Set                        | The proportion of sediment within different size classes distributed         |
|          |                                       | uniformly across the model domain at the start of each simulation.           |
| (3) MNR  | Manning's n Roughness                 | A coefficient related to how much resistance the land surface presents to    |
|          |                                       | flows.                                                                       |
| (4) TOPN | m value used by TOPMODEL              | A parameter that controls the flashiness of hydrograph response to rainfall. |
| (5) VEG  | Vegetation Critical Shear Stress (Pa) | A shear stress threshold above which vegetation is removed from the land     |
|          |                                       | surface by flows.                                                            |
| (6) MAT  | Grass Maturity Rate (yr)              | The amount of time it takes new vegetation to grow to full size.             |

**Appendix A - Description of Parameters**

| (7) MEL | Max Erode Limit (m) |
|---------|---------------------|
|---------|---------------------|

A limit to the amount of sediment that can be eroded in a cell each time

step used to maintain stability.

I think it is important to stress that it is not surprising that outputs that are totaled over the duration of the model run are not sensitive to the DEM resolution, such as total sediment yield. The area of the catchment has not changed, and neither has the average slope of the catchment (I presume). What is more interesting is the response of the model to change, such as how well flood events are recreated. I don't feel that this is really covered by the application of the Morris Method here.

Whilst the totals of sediment yields varied little over different DEM resolutions, the compositions of those totalled did - i.e, sediment yields became larger and less frequent with greater resolutions. The implication is that if a user just looked at the totals they would be missing an important sensitivity of the model. This is core discussion point in the manuscript and the reason we state the model could give the right answer for the wrong reasons.

It would be a useful next step to assess mode outputs versus observed data (which is not always available) but is out of scope for this study, which is concerned with sensitivities of model behaviours to changes.

There is a typo on line 42, "someone".

Corrected.

Why is bedrock erosion ignored? This was also the case in Skinner et al. (2018).

We did not use a bedrock layer for this test and bedrock erosion is not likely to be a key consideration over the 30 year period simulated here. Included:

"The model has capability for bedrock erosion but the set up for this study does not include a representation of bedrock and is unlikely to be influential over the 30 year operational timeframe used. An initial soil layer is determined globally using the information within the Grain Size set parameter (see Table 1)." - Lines 162-164

**Summary**

Overall, I think this manuscript is highly valuable, if focused to users of CAESAR-Lisflood. The results could be possibly extrapolated to other process-based models, such as LAPSUS, PARALEM (?), but this has not been tested. It could be published with some minor improvement, and act as a starting point for more research. Or with some more thought into the question of the spatial distribution of erosion and deposition could make for a bigger piece of work. I would prefer the latter, hence my choice for "major revisions" however being realistic, I would leave that choice to the authors as the day job can get in the way of big revisions.

I hope these comments are helpful.

**John Armitage**

**IFP Energies Nouvelles, Paris**

Thank you for the comments and they were indeed useful. It is appreciated that the results are possibly limited to users of CAESAR-Lisflood but our primary motivation was to demonstrate how the method can used to help guide a user in key decisions – here being what DEM resolution to use when balancing detail and computational efficiency. This is increasingly important when LEMs, like CAESAR-Lisflood, are being picked up for operational and decision-making purposes and not just for research.

There is tremendous scope for further research and you have identified several interesting and valuable ways that this could be done. Unfortunately, the first author of the work no longer has an academic position and would be unable to perform this work for this manuscript, so our preference would be to publish with the minor revisions as you suggested – the methods and data used are openly available for others to take this forward.

**Reviewer #2**

This is a methodological paper exploring the impact of grid resolution on the CAESARLisflood Landscape Evolution Model. The authors use an earlier developed global sensitivity analysis, referred to as the Morris Method to evaluate the impact of grid resolution compared to user-specified model parameters. They conclude that grid resolution does not change the modeled sediment output flux, but that resolution changes the frequency of events, with fewer big events producing the same output flux. This makes the authors conclude that changing the model resolution might give "the right answer for the wrong reasons".

The topic discussed in this manuscript is of interest to an increasing amount of people applying LEM's. Like the authors discuss, setting the resolution of a LEM is a choice with some implications which are, in some cases, overlooked. Re-applying the Morris exercise (a similar exercise was done for parameter sensitivity in Skinner et al 2018) to test the impact of resolution is useful. Yet, I feel the manuscript should be improved at several points to make it a contribution that adds to the existing literature on LEMs.

**Some points not in order of importance:**

Almost no deposition is occurring in the shown simulations (erosion = 40-60k m3 versus deposition = 0.1-0.4k m3, this is a factor 100-200 difference!). This seems problematic for the goal of this paper since I would expect grid resolution to alter deposition. If almost no deposition occurs, I am not sure this is a proper catchment to test the impact of grid resolution. Given the narrow focus of this paper (only considering CAESAR-Lisflood), at least the full spectrum of erosion and deposition should be covered. I suppose this could be solved by comparing the results in-between smaller and larger catchments.

The motivation of this work was to highlight the impact of user choices on outputs and the potential for using the Morris Method to help guide user decisions in setting up the model. The catchment used is similar to catchments where the CAESAR-Lisflood model is used operationally over the timescales applied in these test (30 years), so the question of how grid resolution impacts the outputs is highly relevant. Therefore, we consider this catchment is an appropriate choice for these

test whilst also acknowledging that we are not presenting a full analysis of the sensitivity of the CAESAR-Lisflood model. Included:

"Whilst LEMs were predominantly developed for experimental purposes, such as to understand broad scale basin behaviour over long time scales, the increasing sophistication of the models, ushered in an era of "second generation" LEMs (Coulthard et al., 2013). This has seen LEMs increasingly used over shorter time frames with smaller grid cell sizes for operational purposes or to support decision making (e.g. Environment Agency, 2021; Feeney et al., 2022; Ramirez et al., 2022; Wong et al., 2021). This operationalisation of LEMs brings with it a need to understand model limitations and uncertainties that may have a bearing over real-world decisions." - Lines 24-29

The point you raise is important though and users need to consider the unique characteristics of their catchment when setting up LEMs. We have been more explicit about the limitations of this study. Included:

**"4.5 Limitations**

Whilst this study has highlighted the influence that grid cell size can have on model behaviour and outputs it is also limited by the narrow set of conditions tested. We have only considered a single LEM and only a single small catchment with its own unique set of conditions. Therefore, whilst many of our findings can reasonably be ported to different models with similar parameterisation and operation we acknowledge that the findings are not generic. Equally, whilst we believe many of our findings are directly relevant to other applications of CEASAR-Lisflood to different catchments, Tin Camp Creek has few depositional zones (e.g. floodplains or alluvial fans) so the majority of eroded sediments leave the catchment entirely. We have deliberately used a short timescale of simulation, just 30 years, that is short by the standards of LEMs but analogous to operational uses of the models to aid decision making. The applicability of the detailed results of this study to other models, catchments, and timescales is unknown but the implications remain, that the sensitivities of models to grid cell resolutions should be understood as part of any study using LEMs. This is particularly pertinent when they are being used operationally." - Lines 419-429

Catchment size (related to previous comment): this is a very small catchment. At high resolution the topological network is strongly altered because of the limited catchment area. In my opinion, a study focusing on one single LEM, should cover a larger domain of catchment areas to study the role of changing grid resolution. Using larger catchment sizes would also resolve the issue on deposition, I assume.

Similarly to the point above, the catchment is small but is analogous to catchments used in realworld applications of the model. It is out of scope of this study to provide a comprehensive appraisal of the sensitivity of CAESAR-Lisflood across all factors – each study and site will be different. We have been more explicit about this limitation, see Section 4.5, Lines 419-429.

From reading the title, I was expecting a study that would cover the impact of model resolution in general. However, this paper focuses on one particular model (CAESARLisflood). It would have been more interesting to see a contribution spanning a wider range of models but focusing on one LEM is acceptable. However, it should be indicated as such in the title. Also, I would like to see a discussion on how these findings can be extrapolated to other LEM's. One more comment regarding the title: it is a little weird to use an abbreviation (DEM) in the title. Why not using 'spatial model resolution' or 'grid resolution' rather than DEM grid cell size? DEM resolution might be controlled by other factors

such as the resolution of the source data the DEM was built from. When you say model grid resolution, you avoid this confusion.

We have changed the title of the manuscript to reflect this. It is now "Testing the sensitivity of the CAESAR-Lisflood Landscape Evolution Model to grid cell size".

From the model description, it is unclear whether bedrock incision is simulated. How is sediment being produced? Is the full model domain supposed to be sediment (transport limited behavior versus detachment limited behavior?) or is there a conversion mechanism to transform bedrock into sediment (fluvial bedrock incision, landslides,...).

CAESAR-Liflood is capable of simulating bedrock and bedrock erosion but we have not used this for this study. Included:

"The model has capability for bedrock erosion but the set up for this study does not include a representation of bedrock and is unlikely to be influential over the 30 year operational timeframe used. An initial soil layer is determined globally using the information within the Grain Size set parameter (see Table 1)." - Lines 162-164

Do you assume an initial soil cover where the sediment is derived from? This might be described in the original CAESAR publications but would be good to summarize here.

Soil cover is uniformly distributed globally using the sediment sizes and proportions provided by the Grain Size Set parameter. This is now explained in Lines 162-164.

Over what timescales do observations hold and how sensitive are they to the initial boundary condition (DEM). More broadly speaking, several LEMs are used to create synthetic landscapes to test specific scenarios. What happens if CAESAR-Lisflood is used to generate synthetic landscapes and how sensitive are these kinds of landscapes to changes in resolution (e.g. in terms of fluvial network).

Our focus here was on an operational timescale (30 years). It is out of scope of this study to assess the sensitivity over longer timescales. We have included this as a limitation of the study, please see Section 4.5, Lines 419-429.

The introduction should be structured better. Now it reads as an enumeration of various studies, but I miss a good story line here. It would be helpful to provide an overviewing first paragraph where the authors summarize what they will discuss and for which types of models. Next discuss these points and clarify what is known from these studies and what the knowledge gaps are. From there move towards the final paragraph outlying what will be done in this paper. Also, this work builds on Skinner et al. It would be useful to provide a summary of their main findings and explain how this manuscript builds on those using various grid resolutions.

We have edited the introduction to reflect these points and included additional contextual information for the motivation of the study in Lines 24-29.

Grid resolution is one thing, numerical methods another. The latter is not mentioned in the manuscript but is critically important regarding grid resolution: some numerical methods will be more sensitive to grid resolution than others. The paper would benefit from details on the numerical implementation of the model as well as details on the temporal properties of the simulations. Numerical methods determine the sensitivity of LEMs to grid resolution. Finite Difference Methods will respond differently to changing resolutions compared to Finite Volume Methods or Finite Element Methods. Moreover, I am wondering how the numerical model advances in time: is a forward difference scheme used or a more complex scheme (e.g. Runge Kutta,...)? Related to that: provide details on the timescale over which this model is run and the timesteps being used. In terms of model performance, as shown and as expected, models run faster at lower resolution. Should be good to discuss whether the model allows parallelization and how that alters performance for various grid sizes.

We consider this information would be superfluous to the manuscript here as our motivation is to highlight that user choices influence model outputs, and to present the Morris Method as a relatively efficient way for users to test model sensitivities to these. Descriptions of the numerical methods in the model are provided in previous papers should readers wish to delve into this detail.

The timescale of the simulation is stated in the manuscript - 30 years (see Lines 245-247). The model uses a variable timestep method and we have used the default min and max timesteps of 1s and 360s. We assessed model performance using number of iterations as a hardware independent proxy for relative time taken.

Figure 1 is hard to interpret before reading the methods section. This figure comes directly from Skinner et al and I see little value in doing so. Rather provide the readers with a synthetic figure overviewing the method and move it to the methods section.

We agree that the figure contributed little to manuscript and have removed it.

One suggestion would be to replace this figure with a synthetic figure that describes 3-5 cases (dots on the graph). For every case, the authors can explain what a high mean versus standard deviation imply, and how it should be interpreted.

We decided that a figure was not required and have removed it.

Would be good to explain the different transport formulas and how they behave differently. It might be explained elsewhere but knowing how they work is critical to understand what is going on in this study so please summarize.

This granularity in detail is not required for the purposes of this work – we want to demonstrate the impact of user choices has on the outputs, not provide a detailed analysis of each and every component of those choices.

Stream network analysis: it is obvious that you lose details when coarsening a DEM, no modelling is needed to show that. What would be an interesting exercise is to run a synthetic LEM using a variable resolution and to check how the stream network comes out. Checking the order might be

one metric to look at, but you could also consider evaluating the drainage density. This exercise could also be tested on drainage basins of various sizes (see before).

This would be an interesting test and an area of future work perhaps but out of scope for this work.

**Fewer yet more erosive events: is there a process-based explanation for this?**

This is explained in the discussion as related to the loss of detail in the stream network, making areas of the network less likely to be 'switched on' during flood events, but when they do the flows are larger and more erosive. See lines 353-356.

Generally, with decreasing resolution, gradient decreases. Intuitively, I would expect this to result in decreasing erosion rates for single erosive events rather than increasing events. Some background on how erosion works in CAESAR-Lisflood might clarify this.

This is explained in lines 353-356. When flows do occur at coarser resolutions they are larger. The non-linear relationship between flow velocity and sediment transport means these higher flows are having a disproportionate impact on sediment yields.

Are there thresholds involved in the erosion mechanisms implemented?

Yes.

Thank you for your comments, they were helpful and will contribute to a better manuscript. We agree that more tests on a wider range of catchment sizes and types would be useful but consider them to be out of scope with the main motivation of the study.

In line comments provided in an attached PDF by reviewer #2 have been addressed as appropriate throughout the revised manuscript.

---

## Author Response (AR2)

Dear Chris and Tom,

thanks for your replies to the two referrees and the revised manuscript. I agree with many of your replies to requests for further analyses such as including a simpler catchment, tests on a wider range of catchment sizes and types, and variable initial conditions. Most of these requests would be outside the scope of this study. Still, I'd also like to stress that scope of implications of the study should be beyond the use of CAESAR and a 30 year run in a specific catchment. In a revised manuscript, I'd hope to see which of the findings are general and which are specific to this model setup.

Reviewer 2 also asks for a better explanation or summary of the different transport forumulas and how they behave differently because they are critical to understand what is going on in this study. You reply that this granularity in detail is not required for this work. In a revised manuscript, I'd ask you to take the comments and suggestions more seriously, and provide these information concisely in a revised manuscript. Readers may not ask for a full derivation of the formulas, but a brief explanation would increase the readability of the paper. The same is true for the numerical methods used by CAESAR. I think that the reviewer makes a valid point that numerical methods differently deal with changes in spatial resolution.

All in all, this is a highly interesting paper and I thank you for submitting to ESURF. I hope to receive your revisions soon.

With best regards,

Wolfgang

Dear Wolfgang – thank you for the editorial steer with the manuscript.

For your first point (paragraph one) we have merged the final limitations section into an implications and limitations section – and added six lines where we expand about which findings are relevant just for CL and more importantly what the findings mean for wider use and application of LEMs.

For the Second section, we have added to the methods section – hopefully covering the issues about the numerical methods and the sediment transport rules used. I've tried to strike a balance here to give a clear prose based description of the salient points of the methods and their operation – without including the equations and a description of the terms etc.. the references are there for the reader to find this out. I hope this is appropriate – if needed I can expand this and drop in some equations and further expand.

All the best,

Tom and Chris.

---

## Author Response (AR3)

Dear Chris and Tom,

thanks for submitting your revised manuscript. It is really a pleasure to read the manuscript, and I particularly like how you generalize your findings in the last paragraph of the discussion. Accordingly, I have only a few minor comments that you should address before the manuscript is ready for publication.

All the best,
Wolfgang

Thank you Wolfgang for reviewing the manuscript and your positive words. We have made edits in line with your comments, below:

8: consider rephrasing: the surface of the Earth and other planets.

Have made this change as suggested.

Fig. 1: This map is - to put it mildly - a bit basic. Although not very relevant for this study, it would be good to have a map that shows where the site is, perhaps illustrated also by satellite imagery that enables readers to better grasp the environmental setting of the site. Also consider sparing white space.

Have included more location information and satellite imagery in a new Figure 1.

Appendix B: This figure is not very helpful and I'd consider either deleting it or creating hillshades which enable readers to much better appreciate the differences in land surface representation by the differently resolved DEMs.

We have removed this Appendix B and updated references throughout.

Finally, I'd like to encourage you to make the simulations (and pertinent information) publicly available (e.g., via a repository such as Zenodo). In fact, this is also true for the many hundreds of model runs which were conducted as part of the study published in GMD a couple of years ago. I think that in terms of inter-model comparison, a lot can be done with this data and it would be great to offer access to it also for others.

We have uploaded the test and output files to Zenodo and they can be found here: DOI:10.5281/zenodo.7908491

We accept your request to similarly provide the information for the data from Skinner et al 2018 and will look to make this available in the near future.